# Forest-of-Thought: Scaling Test-Time Compute for Enhancing LLM Reasoning

**Zhenni Bi** [† 1]   **Kai Han** [† 1]   **Chuanjian Liu** [1]   **Yehui Tang** [* 1]   **Yunhe Wang** [* 1]

## Abstract

Large Language Models (LLMs) have demonstrated remarkable abilities across various language tasks, but solving complex reasoning problems remains a significant challenge. While existing methods, such as Chain-of-Thought (CoT) and Tree-of-Thought (ToT), enhance reasoning by decomposing problems or structuring prompts, they typically perform a single pass of reasoning and may fail to revisit flawed paths, compromising accuracy. To address this limitation, we propose a novel reasoning framework called Forest-of-Thought (FoT), which integrates multiple reasoning trees to leverage collective decision-making for solving complex logical problems. FoT employs sparse activation strategies to select the most relevant reasoning paths, improving both efficiency and accuracy. Additionally, we introduce a dynamic self-correction strategy that enables real-time error correction, along with consensus-guided decision-making strategies to optimize both correctness and computational resources. Experimental results demonstrate that the FoT framework, combined with these strategies, significantly enhances the reasoning capabilities of LLMs, enabling them to solve complex tasks with greater precision and efficiency. Code will be available at https://github.com/iamhankai/Forest-of-Thought.

## 1. Introduction

Large Language Models (LLMs) have revolutionized natural language processing by demonstrating remarkable capabilities across a wide range of language tasks. By leveraging vast datasets and complex architectures, LLMs such as ChatGPT (Kojima et al., 2022; Achiam et al., 2023) and LLaMA (Touvron et al., 2023) can generate coherent essays, answer complex questions, and engage in multi-turn dialogues with human-like fluency. These models excel at tasks requiring not only linguistic understanding but also basic reasoning, such as translating text, summarizing lengthy documents, and generating code from plain language instructions. The versatility and adaptability of LLMs have made them invaluable tools in both industry and research, opening up new avenues for solving general-purpose problems.

Enabling LLMs to successfully solve complex reasoning problems remains a challenge. A series of works have been proposed to introduce more inference during testing based on a well-trained LLM (Wei et al., 2022; Yao et al., 2024; Snell et al., 2024; OpenAI, 2024). Chain-of-Thought (CoT) (Wei et al., 2022) provides a few chain-of-thought demonstrations in prompting as exemplars to enhance the reasoning abilities of LLMs. Tree-of-Thought (ToT) (Yao et al., 2024) allows language models to explore multiple reasoning paths and self-evaluate to make more globally informed decisions. Graph-of-Thought (GoT) (Besta et al., 2024b) advances LLM prompting by structuring information as a graph of interconnected "thoughts", enabling synergistic reasoning and feedback loops.

These methods perform reasoning by using richer prompts or by decomposing a complex problem into several simpler sub-problems. However, they typically perform only a single, complete reasoning pass on the problem, which does not guarantee that the problem will be solved correctly. For example, in a complex mathematical word problem, a Tree-of-Thought approach might decompose the problem into smaller steps, such as isolating terms or simplifying expressions. However, while breaking down the problem, it may still overlook critical details or make errors in intermediate steps, leading to an incorrect final answer. Once it completes a single reasoning path, it typically does not revisit other possible approaches if the initial path is flawed. This lack of re-evaluation can result in a solution that fails to address the full complexity of the problem, as alternative paths are often prematurely abandoned and left unexplored, thereby compromising accuracy. In contrast, humans tend to repeatedly reflect and verify from different perspectives when dealing with complex problems, which allows them to truly solve the problem and provide answers with higher

---

[*]Equal contribution [1]Huawei Noah's Ark Lab. Correspondence to: Yehui Tang <yehui.tang@huawei.com>, Yunhe Wang <Yunhe.Wang@huawei.com>.

*Proceedings of the $42^{nd}$ International Conference on Machine Learning*, Vancouver, Canada. PMLR 267, 2025. Copyright 2025 by the author(s).

accuracy.

In this paper, we propose a new reasoning framework called Forest-of-Thought (FoT) to scale up test-time computation and enhance the reasoning abilities of LLMs, as shown in Figure 1. FoT integrates multiple reasoning trees to leverage the advantages of collective decision-making for handling complex logical reasoning tasks. By utilizing sparse activation strategies, we select the most relevant reasoning paths for each tree, thereby improving both the efficiency and accuracy of the model. To further enhance the reasoning process, we introduce a dynamic self-correction strategy, which enables the model to automatically identify and correct errors during reasoning, leveraging both real-time corrections and historical learning. Additionally, we incorporate consensus-guided decision-making strategies to optimize both correctness and computational resource usage, ensuring that the model only continues the reasoning process when necessary. Our experiments demonstrate that the proposed FoT framework, combined with these strategies, significantly improves the reasoning performance of LLMs, enabling them to solve complex tasks with greater precision and efficiency.

## 2. Related Works

### 2.1. XoT reasoning

Starting with Chain-of-Thought (CoT), XoT reasoning techniques (e.g., ToT and GoT) emerged as important methods for enhancing the reasoning abilities of LLMs, leading to the development of a series of XoT reasoning algorithms.

**Chain-of-Thought (CoT) (Wei et al., 2022)** decomposes a problem into a series of intermediate steps, each providing a portion of the information necessary for the final answer. This approach mimics human problem-solving strategies, involving step-by-step reasoning to reach a conclusion. However, while CoT performs well in many tasks, it has limitations when applied to complex mathematical and logical problems. These types of problems often require multidimensional, non-linear thinking rather than just sequential reasoning. Despite numerous subsequent studies aimed at improving CoT, such as Zero-Shot-CoT (Kojima et al., 2023), Self-Consistency with CoT (CoT-SC) (Wang et al., 2023), Auto-CoT (Zhang et al., 2022), VerifyCoT (Zhao et al., 2023), and CoF-CoT (Nguyen et al., 2023), further exploration and optimization are still needed to tackle highly complex tasks.

**Least-to-Most Prompting (LtM) (Zhou et al., 2023)** guides the model through a step-by-step process, progressively assisting it in constructing a solution, in contrast to methods like CoT, which attempt to solve complex problems directly. This approach effectively mitigates the reasoning errors that often occur when attempting to solve complex

problems all at once. By decomposing complex problems into simpler tasks, Program of Thought (PoT) (Chen et al., 2023), Chain of Code (CoC) (Li et al., 2024), and Buffer of Thought (BoT) (Yang et al., 2024) transform the process into a set of programmatic steps, using variable names to convey semantic information. In contrast, the Algorithm of Thought (AoT) (Sel et al., 2024) method seeks to integrate these steps into a single prompt, enabling LLMs to learn how to break down problems, generate solutions, assess their feasibility, and determine the next step in the search process. This approach reduces token consumption and improves efficiency.

**Tree-of-Thought (ToT) (Yao et al., 2024)** constructs a tree structure to explore various possible choices and their outcomes, where each node represents a decision point, and the edges represent transitions between states. Typically, a depth-first search (DFS) approach is used to explore each branch incrementally. Tree Prompting (Morris et al., 2023) establishes a decision-tree-based prompting system, chaining multiple language model calls together to collaboratively complete a specific task. However, for complex problems, the tree's depth may become very large, leading to an exponential increase in the search space and a higher computational burden. Graph-of-Thought (GoT) (Besta et al., 2024b) extends the ToT framework by introducing an aggregation process. GoT models the reasoning process of LLMs as a graph structure, allowing information units to form arbitrary dependencies, not limited to linear or tree-based arrangements. Through aggregation, GoT consolidates information from multiple paths, enabling complex dynamic path selection and backtracking. Skeleton-of-Thought (SoT) (Ning et al., 2023) reduces generation latency in LLMs by first generating a skeleton outline of the answer, then completing the content in parallel, achieving significant speedups and potential quality improvements across various types of questions.

### 2.2. Monte Carlo Tree Search

Monte Carlo Tree Search (MCTS) (Chaslot et al., 2008), a probability-based search algorithm, has made significant progress across various domains since its introduction in computer Go in 2006. It evaluates nodes through randomized simulations (rollouts) and incrementally builds a local game tree to identify optimal or near-optimal solutions within a limited time. To enhance MCTS performance, researchers have proposed various improvements. Browne et al. (2012) surveyed extensions such as sparse activation, dynamic pruning, parallelization, and distributed computation, which have broadened MCTS applications. Srinivas et al. (2012) introduced UCB1-Tuned, an improved exploration-exploitation strategy suited for high-dimensional spaces. Recently, integrating MCTS with large language models has advanced its use in complex reasoning

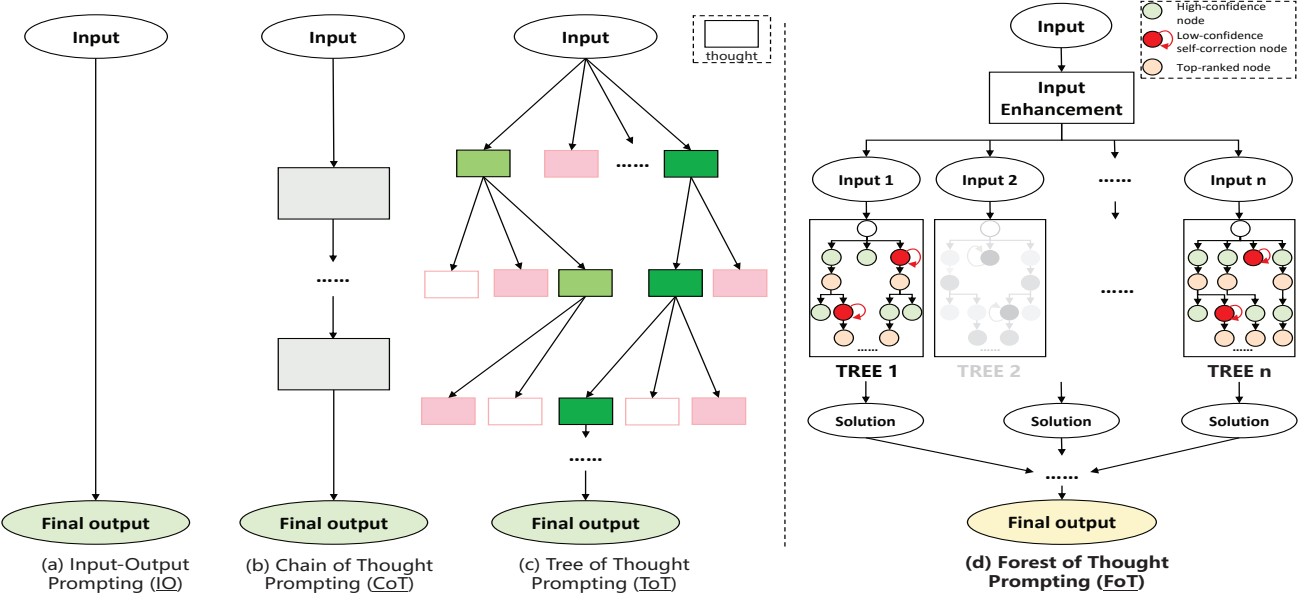

*Figure 1.* Schematic illustration of various LLM reasoning approaches including IO prompting, CoT, ToT and the proposed FoT.

tasks. Everything-of-Thought (Ding et al., 2024) employs a combination of reinforcement learning and MCTS for reasoning. Methods like MCTSr (Zhang et al., 2024) combine MCTS with LLMs to guide decision-making, blending probabilistic search with linguistic reasoning. This hybrid approach enables effective multi-step logic and diverse solution paths.

## 3. Method

By introducing multiple reasoning trees (e.g., ToT (Yao et al., 2024) or MCTSr (Zhang et al., 2024)) for independent decision-making and employing sparse activation strategies to filter the results from key trees, we can construct a Forest of Thought to enhance the reasoning capabilities of LLMs, as shown in Figure 1(d) and Algorithm 1. This strategy leverages collective intelligence to compensate for individual shortcomings, thereby improving the model's ability to reason from multiple perspectives. Our experimental validation demonstrates that integrating the results from multiple reasoning trees through sparse activation strategies indeed enhances the reasoning capabilities of larger models to a significant extent. This finding not only deepens our understanding of model integration techniques but also provides new insights for improving the mathematical reasoning abilities of LLMs.

### 3.1. The FoT Framework

Suppose we have $n$ reasoning trees $T_1, T_2, \cdots, T_n$, each of which approaches the input problem from a different

perspective. The root node of each tree represents the initial state or problem input, and each subsequent node represents an intermediate result or step in the reasoning process. Let the input be $x$. Each reasoning tree will start from the input and produce a result through different reasoning steps:

$$s_i = T_i(\varepsilon(x)), \quad i \in \{1, 2, \ldots, n\} \tag{1}$$

where $\varepsilon(x)$ is a function that enhances $x$ before it is used as input (details in the following paragraph) and $T_i(\cdot)$ represents the reasoning process of the $i$th tree. FoT considers the results of these trees in a sparse activation manner and produces a high-quality response through a decision-making strategy (Sec. 3.3). Additionally, a dynamic self-correction strategy is proposed to enhance the accuracy (Sec. 3.2).

**Sparse Activation.** In the reasoning process of FoT, we aim to improve both computational efficiency and answer quality by selectively activating only the most promising reasoning trees. Instead of exhaustively computing over all trees, sparse activation ensures that inference resources are allocated only to paths that demonstrate strong intermediate reasoning signals.

At a high level, sparse activation works by evaluating the intermediate outputs of each reasoning tree at every layer and filtering out trees that are unlikely to contribute meaningful information to the final decision. This is particularly crucial in multi-step reasoning tasks, where invalid or low-quality intermediate outputs can lead to cascading errors in subsequent steps. By terminating these unpromising paths

early, we save computation and reduce noise in final output aggregation.

For each reasoning tree $T_i$, at every layer, we evaluate all generated nodes and select only the top-scoring candidates (based on a task-specific scoring function, such as model log-likelihood or heuristic correctness). These selected nodes are then split into child nodes to form the next layer. If a tree fails to generate any valid outputs at a given layer—e.g., due to invalid syntax, semantic contradictions, or domain-specific constraints—its expansion is halted prematurely. The tree is marked as inactive, and it does not participate in the final answer aggregation.

Formally, the activation indicator $\varphi_i$ for tree $T_i$ is defined as:

$$\varphi_i = \begin{cases} 1, & \text{if } \forall l, \ F(s_l) = \text{valid output}, \\ 0, & \text{otherwise.} \end{cases} \quad (2)$$

Here, $s_l$ is the solution or intermediate output at layer $l$ in tree $T_i$, and $F(s_l)$ is a validation function that determines whether the output is logically or semantically valid. The tree is considered *active* only if all its intermediate layers produce valid outputs; otherwise, it is pruned from the final ensemble.

Sparse activation therefore functions as both a filtering mechanism and a computational budget control strategy. It reduces inference latency and improves accuracy by:

- Pruning low-quality or invalid reasoning paths before they reach full depth.

- Allowing the model to concentrate its computation on a smaller number of high-confidence paths.

- Enhancing the diversity and quality of the final answer pool by removing noisy or erroneous candidates.

In practice, we find that sparse activation improves the trade-off between reasoning depth and breadth, leading to more robust and interpretable outputs with lower overall cost.

**Input Data Augmentation.** When faced with complex problems, our cognitive process typically shifts from rapid, intuitive "fast thinking" to deeper, more systematic "slow thinking" (Evans, 1984; Kahneman, 2011). This transition not only helps us address immediate challenges but also draws on the relevant prior knowledge stored in our brains, enabling us to analyze and solve problems more comprehensively. Slow thinking integrates and evaluates this prior knowledge, allowing us to view problems from multiple perspectives and arrive at more reasonable solutions. Inspired by prior knowledge, we have collected and constructed a knowledge base $\mathcal{B}$ from publicly available datasets to support the model's reasoning process. Additionally, by enhancing the diversity of question-answer generation within

each tree to further improve the model's performance.

$$i_{max} = \arg \max_i \left( Sim \left( x, \mathcal{B} \right) \right) \quad (3)$$
$$x = \mathcal{B}_{i_{max}} \oplus x \quad (4)$$

where $Sim(\cdot)$ represents retrieving the most relevant questions from $\mathcal{B}$. $\oplus$ means the concatenation of two text strings.

The sparse activation strategy improves both efficiency and accuracy by focusing on the most relevant reasoning paths and reducing unnecessary computations. To further optimize reasoning, we apply efficient search with early termination. When a clear target is identified (e.g., Game of 24, Arithmetic Puzzles and Maze Solving), the search halts immediately upon finding a solution. This avoids redundant computations: once a branch matches the ground truth (GT), further exploration of irrelevant paths is stopped, saving resources and speeding up the process. Early termination thus enhances overall efficiency by preventing unnecessary work.

---

**Algorithm 1** Forest of Tree (FoT)

**Require:** Input $x$, LLM $p_\theta$, $n$ reasoning trees $\{T_i()\}$, $i = 1, 2, \cdots, n$;
1: $\mathcal{S}_0 \leftarrow \{\}$
2: **for** $i = 1, \cdots, n$ **do**
3:     obtain result of the $i$-th tree with input enhancement: $s_i, \varphi_i \leftarrow T_i(\varepsilon(x))$;
4:     dynamic self-correction: $s_i' \leftarrow self\text{-}correct(s_i, x)$ (Sec. 3.2);
5:     **if** activator indicator $\varphi_i == 1$ **then**
6:         update result set: $\mathcal{S}_i \leftarrow \mathcal{S}_{i-1} \cup \{s_i'\}$;
7:     **else**
8:         **continue**;
9:     **end if**
10: **end for**
    **Return** Decision Making $CGED(\mathcal{S}_n)$ (Sec. 3.3).

---

### 3.2. Dynamic Self-Correction Strategy

Self-correction is a fundamental cognitive method that humans use to solve complex problems (Simon, 1991; Flower, 1981; Amabile, 1983). Unlike Self-Refine (Madaan et al., 2023), which relies on a fixed number of iterations, our approach dynamically evaluates each reasoning step, specifically by monitoring the predicted logits scores to assess the quality of the reasoning results. When the model's score falls below a predefined threshold, a correction mechanism is automatically triggered to detect and fix errors in a timely manner. This approach enhances the flexibility and adaptability of the reasoning process, as it does not rely on preset iteration counts, but instead adjusts the model's output based on real-time feedback.

Additionally, our method incorporates predefined mathe-

matical rules, further improving the accuracy and reliability of the reasoning process. By embedding these rules into the reasoning framework, the model can immediately correct errors upon detection. For example, in Game 24, the model can verify whether the remaining numbers in the output are derived from the input numbers, enabling quick error detection and correction. This immediate correction mechanism not only improves reasoning quality but also significantly reduces the risk of error propagation. Overall, through dynamic evaluation, real-time feedback, and rule-driven correction mechanisms, our approach enables the model to quickly identify and correct errors, ultimately enhancing its overall performance and robustness.

The process details of the proposed dynamic self-correction strategy are shown in Algorithm 2.

---

**Algorithm 2** Dynamic Self-Correction Strategy

**Require:** Input context $x$, LM $p_\theta$, mathematical rule correction function $F$, priori knowledge sets $\mathcal{B}$;

1: $s_i \leftarrow p_\theta(s_i \mid x)$;
2: $score_i \leftarrow p_\theta(score_i \mid s_i, x)$;
3: **if** $score_i <$ threshold **then**
4:     **if** $F$ is not None **then**
5:         $s'_i \leftarrow F(s_i)$;
6:     **else**
7:         $s'_i \leftarrow p_\theta(s'_i \mid \mathcal{B}, s_i, x)$;
8:     **end if**
9:     update result: $s_i \leftarrow s'_i$;
10: **end if**

---

### 3.3. Decision Making Strategy

To address complex mathematical problems, we designed the Consensus-Guided Expert Decision (CGED) strategy to ensure high accuracy and reliability in the final answers generated by FoT. The CGED approach combines collective intelligence with expert judgment from LLMs, guiding the reasoning process from consensus-based decision-making to expert evaluation.

During the reasoning process of FoT, each activated tree generates the optimal solution for its reasoning path. These solutions then undergo majority consensus voting and expert evaluation to gradually identify the best answer. Specifically, for complex reasoning tasks, if the majority of trees produce inconsistent results, an LLM expert will compare the reasoning processes and outcomes of the different trees, making a final decision based on their professional knowledge and experience. This approach ensures the accuracy of the final result, effectively reduces errors and biases in the reasoning process, and enhances the robustness of the entire framework.

*Table 1.* Ablation experiment of FoT with self-correction, input enhancement, and sparse activation.

| Method | Self-Correction | Input Enhancement | Sparse Activation | Acc. [%] | LLM invoked |
|---|---|---|---|---|---|
| FoT | | | | 10.58 | 32.32 |
| FoT | ✓ | | | 60.24 | 32.32 |
| FoT | ✓ | ✓ | | 77.98 | 32.32 |
| FoT | ✓ | ✓ | ✓ | 77.98 | 26.99 |

## 4. Experiments

We evaluate the proposed FoT method on the widely-used LLM reasoning benchmarks including Game of 24, GSM8K and MATH.

### 4.1. Experimental setups

In this experimental section, we explore FoT methods based on ToT and MCTSr, conducting experiments across multiple LLMs and datasets. For the Game of 24 (Yao et al., 2024), our FoT is built using ToT as the reasoning tree. In addition to the ToT-based FoT, we developed an MCTSr-based FoT to address mathematical problems, including those from the GSM8K (Cobbe et al., 2021a) and MATH (Hendrycks et al., 2021b) benchmarks. Furthermore, we extend our method to models such as Llama3-8B-Instruct (at Meta, 2024), Mistral-7B (Jiang et al., 2023), and GLM-4-9B (GLM et al., 2024), testing it on the GSM8K benchmark to assess its generalization.

### 4.2. Game of 24

The Game of 24, from 4nums.com, involves constructing an arithmetic expression using each of the four given numbers exactly once, such that the expression evaluates to 24. We removed the duplicate and unsolvable problems, leaving 95 problems as the test set. In our experiment, we set the sampling temperature to 0.95. We also conducted a series of ablation experiments based on Llama3-8B-Instruct, as shown in Table 1.

**Results.** Table 1 presents an ablation experiment comparing the performance of FoT and ToT, highlighting the impact of different components on reasoning accuracy. The experiment starts with the BoN method, which directly uses the ToT framework without input enhancement, sparse activation, or self-correction. Under this configuration, the accuracy is relatively low, at 10.58%, as the model relies solely on the base ToT approach, without any optimizations or adjustments to enhance its reasoning capabilities. In contrast, by introducing self-correction, FoT shows a significant improvement, achieving an accuracy of 60.24%. The self-correction mechanism plays a pivotal role in enhancing the reasoning process. It allows real-time adjustments to the current reasoning path, ensuring errors are corrected promptly

as they are detected. This reduces the accumulation of errors that can otherwise arise when later reasoning paths depend on potentially flawed earlier steps. Consequently, FoT with self-correction delivers more accurate and reliable reasoning, especially in complex problem-solving scenarios. Further improvements are observed when input enhancement is introduced alongside self-correction, resulting in the FoT with both features. This configuration boosts the accuracy to 77.98%, highlighting the importance of enriching the model's inputs to provide broader perspectives, which in turn supports more robust reasoning. Finally, when sparse activation is incorporated into the configuration, there is a significant reduction in the number of LLM invocations, from 32.32 to 26.99. This not only improves computational efficiency but also ensures that the model maintains its highest accuracy yet, demonstrating the power of combining self-correction, input enhancement, and sparse activation.

In Figure 5, we compare the computational cost and performance of the extended ToT and FoT methods. In the experiment, we progressively increase the number of selectable leaf nodes in ToT (b = 2, 4, 8, 16, 32) to allow for greater diversity in potential reasoning outcomes. The experimental results show that simply increasing the number of leaf nodes in ToT leads to a gradual improvement in accuracy, though the gains become more modest as the number of nodes increases. However, when the number of leaf nodes per layer was increased to 32, the results showed no significant improvement. This suggests that the system reached a reasoning bottleneck, where further expansion of the leaf nodes did not result in substantial performance gains. Beyond a certain threshold, increasing the number of available reasoning paths appears to offer diminishing returns, likely due to the inefficiencies of excessive node expansion. In contrast, the FoT method, which incorporates techniques such as self-correction, input enhancement, and sparse activation, demonstrates a much more significant improvement in accuracy. These findings highlight that while expanding the structure of ToT provides only limited performance gains, the integration of additional features in FoT leads to a more substantial enhancement in model performance.

### 4.3. GSM8K Benchmark

In addition to the Game of 24, we evaluated the benefits of integrating multiple methods into the FoT framework using the GSM8K (Cobbe et al., 2021b) dataset.

**Results.** As shown in Figure 2, we constructed forests using various methods, including Zero-Shot-CoT, MCTSr with one turn, MCTSr with 4-rollouts, and MCTSr with 8-rollouts. The experimental results demonstrate that as the number of trees in the forest increases, the advantages of the multi-method forest approach become more pronounced.

*Table 2.* Performance Comparison on Game of 24: Our method, with 8 activated subtrees, achieved the highest average ranking across different inference frameworks.

| Method | LLM invoked | Success |
|---|---|---|
| IO | 1.00 | 10.22% |
| CoT (Kojima et al., 2023) | 1.00 | 4.38% |
| CoT-SC (Wang et al., 2023) | 10.00 | 4.38% |
| GoT (k=1) (Besta et al., 2024a) | 7.00 | 5.26% |
| ToT (b=5) (Yao et al., 2024) | 13.74 | 74.00% |
| BoT (Yang et al., 2024) | 3.00 | 82.40% |
| XoT (w/ 3 r) (Ding et al., 2024) | 1.78 | 85.40% |
| Ours | 25.64 | 96.84% |

Notably, the 4-rollouts MCTSr with 2 trees achieved 3.2 % higher accuracy compared to the 8-rollouts MCTSr. Additionally, it outperformed the 8-rollouts MCTSr with 2 trees, highlighting its distinct advantage. These findings suggest that increasing the diversity of reasoning outcomes has a more significant impact on performance than simply extending the depth of individual trees.

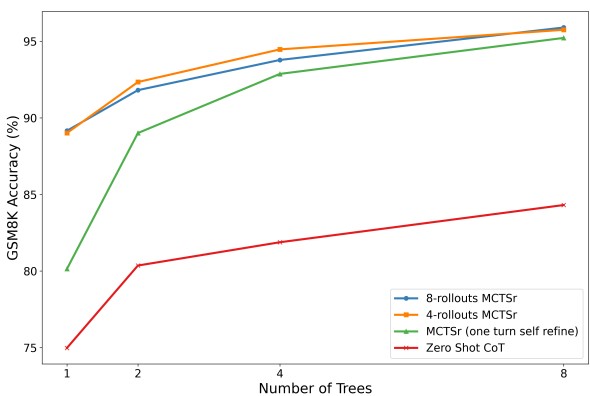

*Figure 2.* Benefit analysis of FoT: the return on wisdom growth. The x-axis represents the number of subtrees in the forest, while the y-axis indicates the accuracy on the GSM8K dataset.

**Scaling laws in FoT across various base models.** In addition to the previously discussed research, we extended our exploration to evaluate the performance of different base models of similar size within the FoT framework. Specifically, we conducted experiments using three models: Mistral-7B, Llama3-8B, and GLM-4-9B. As shown in Figure 3, the results demonstrate a clear scaling law: as the number of activated subtrees in FoT increases, model accuracy improves significantly. This scaling behavior aligns with theoretical expectations, which suggest that activating additional subtrees enhances the diversity of reasoning paths and enables models to refine their problem-solving capabilities. Each subtree incrementally contributed to the overall performance, collectively boosting the framework's reasoning capacity.

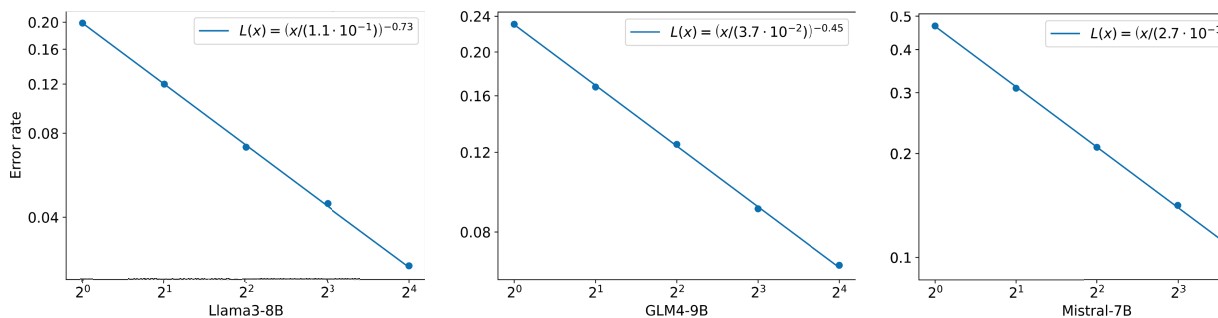

*Figure 3.* Comparative analysis of FoT gains across Llama, Mistral and GLM models. The x-axis represents the number of activated subtrees in FoT, while the y-axis indicates the error rate on the GSM8K dataset. The relationship between the number of activated subtrees in FoT and accuracy has been validated across multiple models, revealing a trend consistent with the scaling test-time comput for enhancing LLM reasoning.

The observed trend suggests a predictable and positive relationship between the number of activated subtrees and accuracy, consistent with scaling law principles. This highlights that the FoT method effectively utilizes computational resources to maximize reasoning accuracy. Moreover, as computational capacity is allocated to activate more subtrees, performance gains exhibit a diminishing, yet consistent, trajectory—showcasing the robustness and scalability of FoT.

## 4.4. MATH Benchmark

This section presents the results of applying the FoT method across various complexity levels on the MATH (Hendrycks et al., 2021a) dataset.

**Results.** The experimental results are shown in Figure 4, where the performance of the FoT method is compared to MCTSr across different difficulty levels in the mathematics dataset. The results clearly demonstrate that the FoT method consistently outperforms MCTSr at every level, from Level 1 to Level 4. Specifically, FoT (n=4) exhibits an impressive and consistent improvement of approximately 10% in performance as the difficulty level increases from Level 1 to Level 4. This steady improvement highlights the robustness and versatility of the FoT method, which is able to adapt effectively to a wide range of problem complexities. At Level 1, which consists of relatively simple problems, both methods perform reasonably well. However, as the difficulty increases, FoT shows a clear advantage, achieving higher accuracy compared to MCTSr at each subsequent level. By Level 4, the most challenging problems in the dataset, the FoT method not only maintains its performance but also demonstrates a significant enhancement in problem-solving capabilities. This consistent improvement underscores the effectiveness of the FoT method in handling problems of varying complexity.

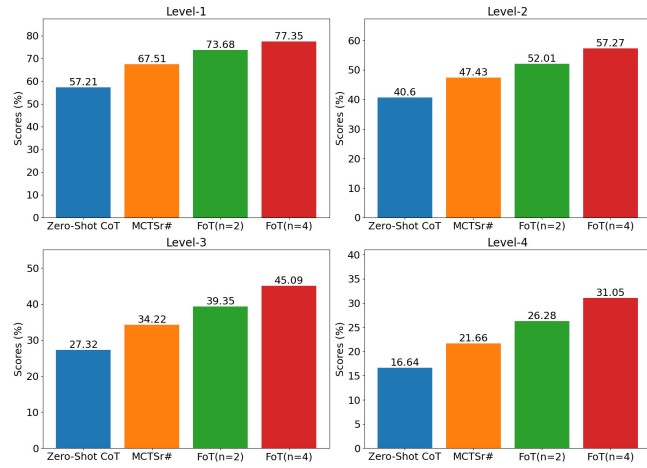

*Figure 4.* FoT demonstrates consistent performance across different levels of the MATH Dataset.

## 4.5. Ablation Studies of Stopping Strategies

In Table 3, we present a detailed comparison of three forest stopping strategies: Majority Vote, Math Expert, and CGED. The results show the performance of these strategies across varying numbers of activated subtrees in the FoT method. When only two subtrees are activated, the accuracy of the CGED strategy is found to be quite similar to both the Majority Vote and Math Expert strategies, indicating that for simpler cases, all three strategies perform comparably. However, as the number of activated subtrees increases, the differences in performance among these strategies become more noticeable. Specifically, when five subtrees are activated, the CGED strategy demonstrates a clear improvement in accuracy, surpassing the Majority Vote and Math Expert strategies by a margin of 2%. This suggests that

the CGED strategy becomes increasingly effective in handling more complex reasoning scenarios, where multiple reasoning paths are activated simultaneously. These findings underscore the superior adaptability and performance of the CGED strategy, particularly in tasks involving a larger number of activated subtrees, where it outperforms the other two strategies in terms of accuracy and overall efficiency.

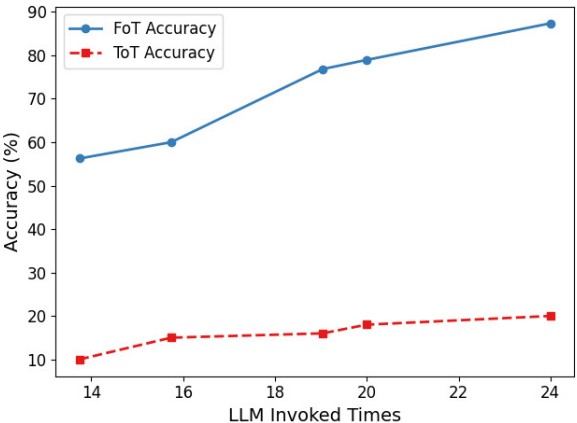

*Figure 5.* On the Game of 24, the performance comparison between FoT and ToT, under the same computational cost.

| Subtrees | Majority Vote | Math Expert | CGED |
|----------|---------------|-------------|------|
| $n$=2 | 80.36 | 80.36 | 80.36 |
| $n$=3 | 81.31 | 81.88 | 84.31 |
| $n$=4 | 84.31 | 82.46 | 84.46 |
| $n$=5 | 83.44 | 83.70 | 85.44 |

*Table 3.* Performance comparison based on the number of trees with three different stopping strategies.

### 4.6. Main Results

As shown in Table 2, the performance comparison on the Game of 24 validates the effectiveness of the FoT in the mathematical and logical reasoning task. The experimental results indicate that the FoT with 8 activated subtrees achieved the highest success rate at 96.84%, significantly outperforming other methods. In contrast, single-step reasoning methods like IO (10.22%) and CoT (4.38%) rely solely on the model's pre-trained capabilities, showing limited performance. Multi-step reasoning techniques, such as ToT (b=5) and BoT, achieved success rates of 74.00% and 82.40%, respectively, demonstrating a notable advantage. Additionally, XoT, after three deployments, reached a success rate of 85.40%. These results clearly indicate that the FoT method, particularly with a higher number of activated subtrees, enhances the model's self-correction capabilities and significantly improves the performance on the Game of 24 while only modestly increasing the computational cost.

*Table 4.* The following summarizes the performance of FoT and other state-of-the-art language models across different levels of mathematical benchmark tests. GSM-level refers to the GSM8K, MATH-level to the MATH500, and Olympiad-level to AIME 2024. Specifically, rStar-Math (Guan et al., 2025) refers to rStar-Math (7B SLM + 7B PPM), and the latter denotes the Pass@1 accuracy achieved when sampling 64 trajectories.

| Model | GSM-level | MATH-level | Olympiad-level |
|-------|-----------|------------|----------------|
| GPT-4o | 92.90 | 76.60 | 9.30 |
| rStar-Math | 95.00 | 89.40 | 46.70 |
| rStar-Math[64] | 95.20 | 90.00 | 53.30 |
| Base Model: Qwen2.5-Math-7B-Instruct | | | |
| Qwen2.5-7B | 88.48 | 82.60 | 6.00 |
| FoT ($n$=2) | 93.33 | 83.02 | 26.67 |
| FoT ($n$=4) | 95.00 | 85.80 | 33.33 |
| FoT ($n$=8) | 96.89 | 86.20 | 46.67 |
| Base Model: QwQ-32B-preview | | | |
| QwQ | 95.30 | 90.60 | 50.00 |
| FoT ($n$=4) | **97.33** | **91.20** | **53.33** |

The results presented in Table 4 highlight the performance of the FoT method across various datasets, including GSM8K, MATH-500 (Lightman et al., 2023), and AIME 2024 (AIMO, 2024). In the GSM8K dataset, FoT with 4 activated subtrees (FoT ($n$=4)) based on QwQ achieved the accuracy of 97.33%, outperforming rStar-Math (95.20%) and GPT-4o (92.90%). FoT ($n$=4) with QwQ achieved a strong performance with an accuracy of 91.20%, surpassing rStar-Math[64] (90.00%). On AIME 2024, FoT showed significant improvements, reaching 53.33%, which is much higher than rStar-Math at 6.66%. These results underscore the effectiveness of FoT in enhancing model performance, particularly as the number of activated subtrees increases, demonstrating its potential to improve reasoning accuracy across a range of mathematical problem-solving tasks.

## 5. Conclusion

This paper introduces Forest of Thought, a novel framework that significantly enhances the reasoning capabilities of large language models by combining multi-path exploration and dynamic path activation. FoT builds upon and unifies concepts from Tree of Thought and Monte Carlo Tree Search, enabling robust, diverse, and efficient reasoning without requiring backpropagation or fine-tuning. It selectively activates the most promising reasoning paths through a sparse activation mechanism, which improves computational efficiency while maintaining high accuracy. FoT is particularly effective in solving complex tasks, offering both performance and generalization benefits. Additionally, the paper provides a systematic analysis of the scaling relationship between reasoning time and accuracy, offering a theoretical foundation for reasoning optimization in LLMs.

## Acknowledgements

We thank the anonymous reviewers for their insightful and constructive feedback, which greatly helped improve this work. We also gratefully acknowledge the support provided by MindSpore, CANN (Compute Architecture for Neural Networks), and the Ascend AI Processor used in this research.

## Impact Statement

The FoT framework introduces a novel and highly efficient approach to improving the reasoning capabilities of LLMs. By leveraging multiple reasoning trees, dynamic self-correction strategies, and consensus-based decision-making, FoT enables LLMs to solve complex problems with higher precision and reduced computational cost. The key innovation lies in the adaptive reasoning process that allows the model to explore multiple paths, detect and correct errors in real-time, and efficiently narrow down the search space. This approach has significant implications for fields that require complex decision-making, such as mathematics, logic, and AI-driven problem solving, leading to more robust and reliable AI systems.

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

# A. Example Prompts - Game of 24

In the 24-point game, we have designed three types of prompts to guide the problem-solving process:

1. **Propose Prompt**: This prompt is intended to assist players in breaking down the original problem in the first step. It guides players to select two numbers from the given four for calculation, thereby simplifying the problem to the remaining two unused numbers.

2. **Step-2 Prompt**: Based on the result of the first step, this prompt further guides players to choose two numbers from the remaining three for calculation, ultimately simplifying the problem to just one number left.

3. **Value Prompt**: This prompt is used to score each step's response, ensuring that each solution is both reasonable and efficient.

*Table 5.* Game of 24 Propose Prompt

| **Propose Prompt**: |
|---|
| Let's play a game called 24. You'll be given four integers, and your objective is to use each number only once, combined with any of the four arithmetic operations (addition, subtraction, multiplication, and division) and parentheses, to achieve a total of 24. |
| Provide four integers as input. Randomly pick 2 full permutations of the 4 input numbers, perform all four basic arithmetic operations (addition, subtraction, multiplication and division), and list the results and remaining integers after each operation. |
| See <Examples> See Table 8 (Propose Prompt) </Examples> |
| Input: '4 5 6 10' |
| Possible next steps: |

| **Response**: |
|---|
| 4 + 5 = 9 (left: 6 10 9), |
| 10 - 4 = 6 (left: 6 5 6), |
| 5 - 6 = -1 (left: -1 4 10), |
| 4 * 6 = 24 (left: 24 5 10), |
| 10 / 5 = 2 (left: 2 4 6), |
| 4 - 10 = -6 (left: -6 5 6), |
| 5 - 4 = 1 (left: 1 6 10), |
| 10 - 5 = 5 (left: 5 4 6) |

# B. Example Prompts - GSM8K and MATH

We present the prompts utilized for the GSM8K task, since the prompts for the MATH task are essentially the same, with only minor variations in how answers are extracted. In Table 10, we present the results of incorporating a self-correction prompt mechanism into the reasoning process.

*Table 6.* Game of 24 Value Prompt

| **Value Prompt**: |
|---|
| Evaluate if given numbers can reach 24 (sure/likely/impossible). |
| See <Examples> See Table 8 </Examples> |
| Input: '4 + 5 = 9 (left: 6 10 9)' |
| **Response**: |
| impossible |
| **Value Prompt**: |
| Evaluate if given numbers can reach 24 (sure/likely/impossible). |
| See <Examples> See Table 8 </Examples> |
| Input: '4 - 10 = -6 (left: -6 5 6)' |
| **Response**: |
| sure |

*Table 7.* Game of 24 Step-2 Prompt

| **Step-2 Prompt**: |
|---|
| Provide three integers as input. Randomly pick 2 full permutations of the 3 input numbers, perform all four basic arithmetic operations (addition, subtraction, multiplication and division), and list the results and remaining integers after each operation. |
| See <Examples> See Table 8 (Step-2 Prompt) </Examples> |
| Input: -6 5 6 |
| Possible next steps: |

| **Response**: |
|---|
| -6 + 5 = -1 (left: 5 -1) |
| 5 * 6 = 30 (left: 6 30) |
| -6 / 5 = -1.2 (left: 5 -1.2) |
| 6 - 5 = 1 (left: 1 5) |
| 5 - 6 = -1 (left: -1 5) |

This mechanism is designed to enhance the quality of the answers generated by the model. The self-correction prompt operates in two stages: first, it assigns a score to the generated answer based on predefined evaluation criteria, such as logical consistency, accuracy, and relevance. Then, depending on the score, the model is prompted to revise or correct its answer if the evaluation indicates potential flaws or suboptimal reasoning.

By iteratively refining its output, the model learns to identify errors and generate more accurate and coherent responses. This approach not only improves the reliability of the results but also provides a structured framework for integrating feedback into the reasoning process. The experiments summarized in Table 10 demonstrate that the self-correction prompt significantly enhances the model's performance across various tasks, reducing error rates and improving overall answer quality. This highlights the potential of

*Table 8.* Game of 24 problem decomposition example.

**Propose Prompt**

<Examples >
Input: 2 8 8 14
Possible next steps:
2 + 8 = 10 (left: 8 10 14)
8 / 2 = 4 (left: 4 8 14)
14 + 2 = 16 (left: 8 8 16)
2 * 8 = 16 (left: 8 14 16)
8 - 2 = 6 (left: 6 8 14)
2 - 8 = -6 (left: -6 8 14)
14 - 8 = 6 (left: 2 6 8)
14 / 2 = 7 (left: 7 8 8)
14 - 2 = 12 (left: 8 8 12)
2 * 14 = 28 (left: 8 8 28)
</Examples>

**Step-2 Prompt**

<Examples >
Input: 2 8 8
Possible next steps:
2 + 8 = 10 (left: 8 10)
8 / 2 = 4 (left: 4 8)
2 * 8 = 16 (left: 8 16)
8 - 2 = 6 (left: 6 8)
2 - 8 = -6 (left: -6 8)
</Examples>

**Algebra Check**

Input: 8 10
Output:
8 + 10 = 18
8 - 10 = -2
8 * 10 = 80
8 / 10 = 0.8
impossible

*Table 9.* Game of 24 value examples.

**Value Prompt**

<Examples >
Input: 4 10 30
(30 + 4) - 10 = 24
sure
Input: 4 9 11
9 + 11 + 4 = 20 + 4 = 24
sure
Input: 5 7 8
5 + 7 + 8 = 12 + 8 = 20
(8 - 5) * 7 = 3 * 7 = 21
I cannot obtain 24 now, but numbers are within a reasonable range
likely
Input: 5 6 6
5 + 6 + 6 = 17
(6 - 5) * 6 = 1 * 6 = 6
I cannot obtain 24 now, but numbers are within a reasonable range
likely
Input: 10 10 11
10 + 10 + 11 = 31
(11 - 10) * 10 = 10
10 10 10 are all too big
impossible
Input: 1 3 3
1 * 3 * 3 = 9
(1 + 3) * 3 = 12
1 3 3 are all too small
impossible
Input: -1 4 7
(-1 + 7) * 4 = 24
sure
Input: 4 10 30
30 + 4 - 10 = 24
sure
</Examples>

self-correction as a powerful tool for boosting reasoning robustness in complex scenarios.

In Table 11, we introduce a prompt mechanism designed for expert selection, aimed at further refining the model's outputs in scenarios involving multiple plausible answers. When multiple answers align with majority consistency, a mathematics expert evaluates these responses based on domain-specific knowledge and selects the most appropriate or optimal answer.

This approach leverages expert judgment to ensure the final selected answer not only adheres to logical consistency but also aligns with mathematical rigor and best practices. By integrating expert selection into the prompt design, we enhance the reliability and precision of the model's reasoning process.

## C. Evaluating the Scaling of FoT Compute Across Different Baseline Methods.

Tabel 6 demonstrate a clear trend: as the number of activated subtrees grows, the error rate decreases, reflecting the enhanced reasoning capability and robustness of the FoT approach. The scaling law comparison across various methods underscores the efficiency of FoT in utilizing additional computational resources, particularly when contrasted with other models that exhibit diminishing returns or slower error reduction as complexity increases. This suggests that the FoT framework scales more effectively with increased subtree activation, making it a powerful tool for addressing challenging reasoning tasks like those in GSM8K.

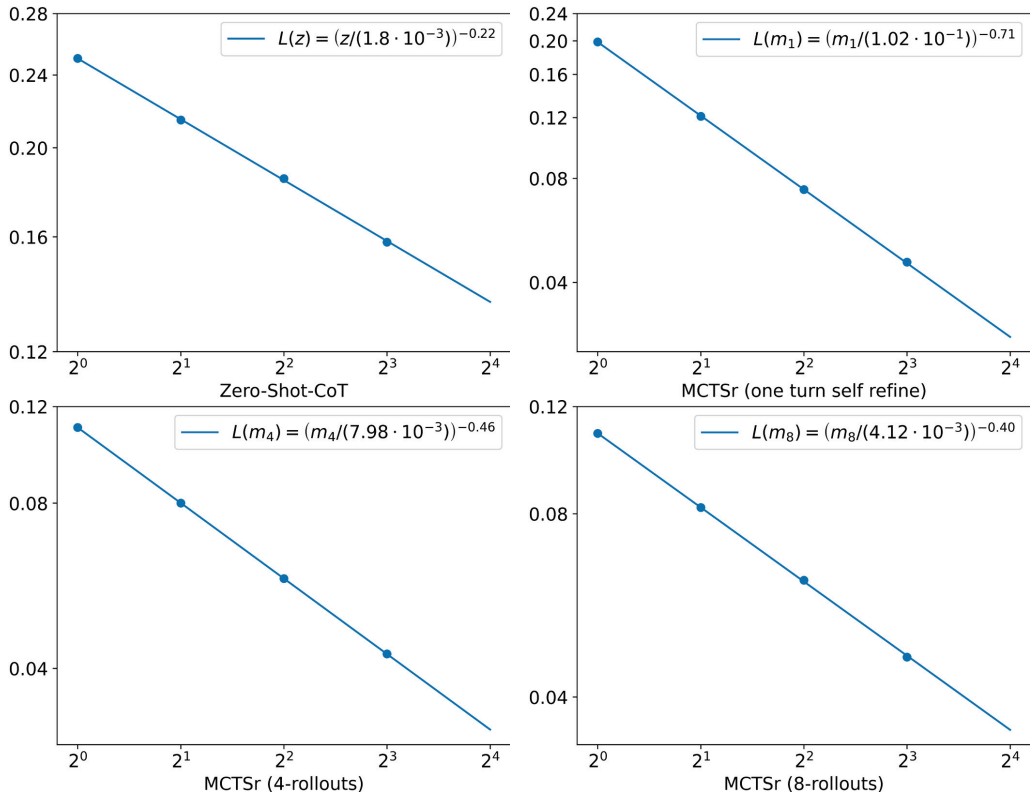

*Figure 6.* The comparison of the scaling laws for FoT across different methods (eg.zero-shot-cot, MCTSr (one-turn-self-refine), 4-rollouts MCTSr, 8-rollouts MCTSr) is presented, with the x-axis representing the number of activated subtrees within the FoT framework and the y-axis indicating the error rate on the GSM8K benchmark dataset.

*Table 10.* Self-Correction Prompt

**Value Prompt**:
Question: {question}
Answer:{answer}
Analyze this answer Strictly and Critic, point out every flaw for ervery possible imperfect to minus every possible score! You need to be very harsh and mean in calculating grades, and never give full marks to ensure that the marks are authoritative. Output a score between [0,100], ig. from 0 to 100. Response format:[Analyst]...[Score]...

**Self-Correction Prompt**:
"Question: {question}
Please refine the your answer according to your Reflection or Feedback. The response should begin with [reasoning process]...[Verification]... and end with end with <ans_format> Let's think step by step."

*Table 11.* Math Expert Prompt

**Math Expert Prompt**: "You are a highly specialized mathematics expert, proficient in solving mathematical problems, and always able to select the most accurate answer from the given options.
**Question:** {question}
**Answers:** {answers_list}
Which of the following answers is the most accurate? The response should begin with <ans_format>."

# D. Evaluating multiple FoT decision-making methods.

Figure 12 compares different decision-making strategies used in FoT when multiple reasoning trees produce conflicting answers. The "Random" strategy selects an answer randomly from all candidates, while the "Score" strategy chooses the answer with the highest confidence score. Experimental results on the GSM8K benchmark show that FoT's reflective decision-making achieves higher accuracy than both baselines, highlighting the effectiveness of leveraging internal reflection over simple voting or scoring.

*Table 12.* FoT decision-making comparison.

| Method | Accuracy |
|--------|----------|
| Random | 77.73 |
| Score | 77.86 |
| CGED | **78.62** |

# E. Evaluating the threshold value in dynamic self-correction.

In Table 13, we evaluate the impact of the self-correction mechanism by conducting experiments at different confidence threshold levels. We systematically adjusted the threshold and observed the model's performance in terms of accuracy. Our findings revealed that when the threshold was set to 0.5, the model achieved significantly better accuracy compared to other threshold values. This suggests that a threshold of 0.5 strikes an optimal balance, enabling the model to identify and correct errors without being too cautious or too lenient in its self-assessment.

*Table 13.* The accuracy performance of the dynamic self-correction strategy with different thresholds on the GSM8K dataset.

| Threshold | Accuracy |
|-----------|----------|
| 0.3 | 87.34 |
| 0.4 | 88.17 |
| 0.5 | 90.14 |
| 0.6 | 88.02 |

