# OpenReview forum: "Forest-of-Thought: Scaling Test-Time Compute for Enhancing LLM Reasoning"
_ICML.cc/2025/Conference — ICML 2025 poster_

### Official Review · Reviewer_SoHo · 2025-03-05

**Overall Recommendation:** 4

**Summary:**

The paper introduces the Forest-of-Thought framework, a novel approach to enhance LLM reasoning by integrating multiple reasoning trees. The key innovations include sparse activation strategies, dynamic self-correction, and consensus-guided decision-making. Experiments suggest that FoT improves accuracy and efficiency in reasoning tasks like MATH.




## ========update after rebuttal======
Thanks for the authors' responses. My concerns have been well addressed.

**Claims And Evidence:**

The paper claims that FoT’s multi-tree integration improves accuracy in complex reasoning tasks by collective exploration of diverse reasoning paths. These claims are supported by experimental results that FoT achieves higher accuracy (e.g., +10% over others) by aggregating multiple trees.

**Essential References Not Discussed:**

N/A

**Experimental Designs Or Analyses:**

The paper evaluates FoT against other sota training-free reasoning frameworks, including CoT, ToT, and BoT, on diverse tasks such as mathematical reasoning (GSM8K), logical puzzles (Game of 24). Experiments employ different models like llama-3, glm-4 and mistral, with rich ablation studies.

**Methods And Evaluation Criteria:**

The proposed FoT framework improves the reasoning accuracy by integrating multiple reasoning trees, and reduces computational redundancy by pruning low-confidence reasoning paths via sparse activation. This aligns with the paper’s goal of balancing accuracy and efficiency. The evaluation criteria is consistent to the compared methods.

**Other Comments Or Suggestions:**

NA

**Other Strengths And Weaknesses:**

Strengths:
1. The motivation is clear, and the proposed framework makes sense. FoT’s multi-tree scheme and decision-making strategy offer a new perspective on scaling test-time compute, addressing the critical limitation of single-path reasoning in prior methods.
2. The paper has demonstrated the superiority of FoT on rich benchmarks and models.

Weaknesses:
1. Although the paper proposes the sparse activation strategy, scaling to multiple trees might still incur significant overhead. The paper lacks a detailed analysis of cost-accuracy trade-off analysis compared to ToT.
2. Since the proposed framework is training-free, it inherently offers better computational efficiency compared to training-dependent methods like DeepSeek-R1. To further strengthen the contribution, it would be valuable to include experiments exploring whether FoT could integrate with trained reasoning models like R1. I'm not sure whether it works, so I’m quite interested in the results. Thanks.

**Questions For Authors:**

See above.

**Relation To Broader Scientific Literature:**

The method seems to be related to ensemble algorithms in machine learning, but there is much different details and innovations in the LLM field.

**Theoretical Claims:**

This paper does not involve theoretical proofs. Its insight can be supported by the ensemble algorithms.

---

> ### Author Rebuttal · Authors · 2025-03-31
>
> We appreciate the reviewers' careful reading and valuable comments. We believe these constructive feedback will help improve the paper. Below are responses to some specific comments.
>
> ---
>
> **Q1:  Weaknesses1: Detailed analysis of cost-accuracy trade-off compared to ToT.**
>
> **A1:** Thank you for your insightful suggestion. We have included experimental results comparing the cost-accuracy trade-off between FoT and ToT, tested on the v100 GPU:
>
> | Method                | Game24 Accuracy (%) | Cost Time (s) |
> |-----------------------|---------------------|---------------|
> | ToT (b=5)            | 56.3                | 412.2         |
> | ToT (b=8)            | 74.7                | 504.6         |
> | ToT (b=16)           | 75.3                | 567.6         |
> | ToT (b=32)           | 76.8                | 664.5         |
> | FoT (n=2, b=5)       | 77.9                | 571.2         |
> | FoT (n=4, b=5)       | 91.6                | 709.2         |
> | FoT (n=8, b=5)       | 96.8                | 769.2         |
>
> ---
>
> **Q2: Weaknesses 2: Supplementary experiments based on Deepseek-R1.**
>
> **A2:** Thank you for your valuable feedback on this manuscript. We present the results of FoT based on experiments with the DeepSeek-R1-Distill-Qwen-7B model across the GSM8K, AIME2024, and Math500 datasets. Further integrating the R1 model with the FoT framework led to improvements across different datasets: a 6% improvement over the baseline model on GSM8K, a 13.3% improvement on AIME2024, and consistent performance on Math500. These findings show that introducing multiple reasoning trees (n=4) enhances the model's reasoning capability and accuracy in problem-solving tasks.
>
>
> | Method       | **GSM8K** | **AIME 2024** | **MATH500** |
> |--------------|-----------|---------------|-------------|
> | DeepSeek-R1-Distill-Qwen-7B    |    89.6  |         53.3   |     92.8        |
> | FoT (n=4)    |  95.5   | 66.6         |      93.3       |
>
> ---

---

### Official Review · Reviewer_JDic · 2025-03-11

**Overall Recommendation:** 2

**Summary:**

This paper presented the Forest-of-Thought (FoT) to extend the former CoT and ToT via integrating multiple reasoning trees and making majority vote, and to avoid the computation complexity of building numerous trees and further improve the performance, the paper employed a set of sparse activation and self-correction approaches to select more possible paths. Empirical results showed the effectiveness of the proposed FoT.

## update after rebuttal
During the rebuttal and discussion phases, I have read all the responses from the authors carefully, there are still two major concerns remaining:
1. This paper proposed to exchange the efficiency for performance, and such a trade-off leads the FoT to outperform previous CoT-variation with more computing cost, while underperforming the more advanced reasoning LLMs, making its applicability ambiguous.
2. The sparse activation algorithm is quite important for understanding the pruning process of the FoT, and I have asked for the details in all rounds of rebuttal and discussion, but haven't accessed it.

Given the above unaddressed concerns, I prefer to hold my previous overall recommendation (weak reject) for this paper.

**Claims And Evidence:**

The claims made in this work are all supported by the following experimental evidence.

The proposed method and evaluation criteria are reasonable for the research purpose.

**Essential References Not Discussed:**

Some related works had not been discussed in the paper.

1. Scaling LLM Test-Time Compute Optimally can be More Effective than Scaling Model Parameters.

2. Improving LLM Reasoning through Scaling Inference Computation Time with Collaborative Verification.

3. Large Language Monkeys: Scaling Inference Compute with Repeated Sampling.

**Experimental Designs Or Analyses:**

I have checked the experimental designs and the results analysis.

The comparison experiments with CoT, ToT, BoT, and MCTS methods, the experimental settings are fair.

However, the authors didn't clarify the computation cost as FoT test-time scaling up, which might consume more computation resources.

**Methods And Evaluation Criteria:**

The evaluation settings are common in similar research works and match the method.

There were no theoretical claims in the paper.

The experimental results are sound and seem to be reproducible.

I have reviewed all the supplemental materials of this paper.

**Other Comments Or Suggestions:**

1. The authors could offer detailed explanations for the sparse activation strategies, especially how it chooses more plausible paths.

2. This paper could present a simple computation/cost ratio to make the performance comparison more fair and better for illustrating the effectiveness of the proposed strategies.

**Other Strengths And Weaknesses:**

Strengths:

1. Integrating multiple reasoning trees with sparse activation enhances the accuracy and efficiency.

2. The experiments for the proposed method and other baselines demonstrate the effectiveness.

3. The LLMs employed in this work are quite enough, showing the superiority of the FoT.

Weaknesses:

1. Building multiple reasoning trees is essentially a test-time solution search method, demanding lots of computation resources, not applicable for low-computation scenarios.

2. This work didn’t compare the computation cost of FoT and other baselines, making solely an accuracy performance comparison that is less fair.

3. Lacking theoretical justification for the sparse activation strategy.

4. All experiments were conducted on math problem solving tasks, limiting the generality for other tasks such as coding, logic, and reasoning for other subjects.

**Questions For Authors:**

1. How does FoT perform under different numbers of reasoning trees?

2. What if we conduct the FoT on other tasks that consensus among all trees is not accessible?

**Relation To Broader Scientific Literature:**

This work discussed several related works like CoT, ToT, and GoT, as they are quite similar. This work extended the ToT, which is one single reasoning tree, to the forest of multiple reasoning trees, which is an interesting variation.

**Theoretical Claims:**

This paper conducted extensive experiments and analyzed results to prove superiority, but didn’t provide theoretical derivatives.

---

> ### Author Rebuttal · Authors · 2025-03-29
>
> Thank you for your thoughtful comments and valuable suggestions. We appreciate the time spent evaluating our work, and we have carefully considered each of your points. Below are our responses to the concerns and suggestions raised.
>
> ---
>
> **Question 1 & Suggestion 2: FoT's performance varies with different numbers of reasoning trees.**
>
> **A1:** As the number of reasoning trees increases, the model's accuracy improves through evaluating multiple paths, as shown in Figure 3. The accuracy gains follow a scaling law, with improvements slowing down as the number of subtrees increases. We also conducted experiments on the computational costs of different subtrees in the AIME task, using the Deepseek-R1-Distill-Qwen-7B on the v100 GPU. The results show that while accuracy increases with more subtrees, the computational cost does not increase exponentially. Additionally, we included experiments on the Game of 24 task's computational cost, as requested by Reviewer 4's Q1. These experiments demonstrate that FoT sacrifices increased inference time for higher accuracy.
>
> | **Subtree Nums** | **AIME Acc** | **Time (s)** |
> |------------------|--------------|--------------|
> | 1                | 53.33        | 13659.63   |
> | 2                | 56.67        | 25014.82  |
> | 4                | 66.66        | 25894.76  |
>
> ---
>
>
> **Q2: The decision strategy when the results of the subtrees are inconsistent.**
>
> **A2:** As described in Section 3.3, when FoT cannot reach a consensus among the multiple reasoning trees, it synthesizes the results and reflects on the different reasoning paths. This approach improves accuracy by considering multiple possibilities and enhancing robustness, without depending on a single path. Experiments show that, after reconsidering all answers, FoT yields more accurate results than random selection or scoring-based methods.
>
> | **Decision Method**  | **Accuracy**  |
> |----------------------|---------------|
> | Random               | 77.73         |
> | Score                | 77.86         |
> | Re-thinking (CGED)   | **78.62**   |
>
> ---
>
> **Q3: Suggestion 1 & Weakness 3: Provide additional details on the sparse activation strategies.**
>
> **A3:** Sparse activation focuses on activating the most promising paths during inference, rather than exploring all possible reasoning paths. This approach ensures accuracy while reducing computational resource consumption. The strategy improves computational efficiency by focusing on the most likely reasoning paths. The framework scores each path, activating those with higher scores and discarding the rest. We will include more detailed steps in the final version.
>
> **Q4: Weakness 1: The practicality of FoT in low-computation scenarios.**
>
> **A4:** FoT is well-suited for handling problems that involve complex logic and require detailed reasoning. By activating multiple reasoning paths simultaneously, FoT can explore problems from different perspectives and logical levels. This approach is particularly effective for solving complex scenarios that single-path methods fail to cover comprehensively. We appreciate your suggestion and are continually optimizing the framework to enhance the efficiency of thoughtful reasoning, making it suitable for low-computation scenarios.
>
>
>
> **Q5: Weakness 2: Compare the computation cost of FoT and other baselines.**
>
> **A5:** Table 2 compares the performance of FoT with other baselines in terms of accuracy and computational cost (measured by
>  Average LLM calls). In addition, we supplemented experiments in response to Reviewer 2(zTLg)'s Q3, where we conducted multiple rounds of LLM calls for various methods. These results demonstrate that FoT achieves higher accuracy, even with more LLM calls, showcasing its efficiency in producing high-quality reasoning results.
>
>
>
>
>
> **Q6: Weakness 3: Theoretical justification for the sparse activation strategy.**
>
> **A6:** Sparse activation focuses on the most promising paths during inference, rather than exploring all paths. This improves accuracy while reducing computational resources. The framework scores each path, activating the highest-scoring ones and discarding the rest. We will provide more details in the supplementary materials.
>
>
>
>
> **Q7: Weakness 4: Experiments on other subjects.**
>
> **A7:** Thank you for your valuable suggestion. We have conducted additional experiments on FoT with the general task CEVAL and the coding task LiveCode. FoT demonstrated a 6% improvement on the general test set and a noticeable gain on the coding task. We will continue to enhance the framework's adaptability to other subjects in future versions.
>
> | Method|**CEVAL** |**LiveCode**|
> |--------------|-----------|---------------|
> | DeepSeek-R1-Distill-Qwen-7B|89.6|37.6|
> | FoT (N=4)|92.5|38.2 |
>
> ---
>
> **Q8: About essential references not discussed.**
>
> **A8:** We believe it is important to consider these relevant references. We will further explore the methods outlined in them in our future work to enhance our approach.
>
> ---

---

> > ### Comment · Reviewer_JDic · 2025-04-05
> >
> > Thanks for the active response! Some of my concerns were addressed while others are still open.
> >
> > In the time cost comparison of A1, with the subtree increasing from 1 to 2, the inference time almost doubles, while the accuracy improvement does not gain much benefit (3.3 is just 1 more correct answer of 30 questions in AIME24). When the subtree number rises to 4, the improvement is apparent, considering the little improvement between 1 and 2, does it imply that FoT is not a stable algorithm?
> >
> > Although the consensus (majority voting) of multiple paths can enhance the performance, the improvement may come from the vast computation cost as shown in Table 2, in which the FoT took 20 times LLM invoke and only ~10% success rate gain. Additionally, the heavy computing cost prevents the application and reproduction for the following researchers.
> >
> > The sparse activation strategy is critical for explaining the pruning process, it's necessary to introduce the details in the main body and provide related theoretical derivations or proofs to make it convincing.

---

> > > ### Author Response · Authors · 2025-04-08
> > >
> > > **Q1: The stability of the FoT algorithm.**
> > >
> > > **A1:** Thank you for your insightful question. In the AIME task, when the number of subtrees increased from 1 to 2, the accuracy improved from 53.33% to 56.67%, but the inference time increased significantly. With fewer subtrees, the accuracy improvement was limited. However, when the number of subtrees increased to 4, the improvement in accuracy became more pronounced. This suggests that FoT is not inherently unstable; rather, its performance improvements become more evident once a certain threshold is reached. Under the phenomenon of emergence in large models, when the number of subtrees exceeds a threshold, it triggers the emergence of the model's complex reasoning capabilities. As more subtrees are added, the model begins to exhibit stronger reasoning abilities, resulting in significant improvements in both accuracy and decision-making, while keeping the increase in computational cost manageable.
> > >
> > > Additionally, the performance improvement of FoT is also influenced by the complexity of the test task. In response to Reviewer 4's Q1, we observed a notable accuracy improvement in the Game of 24 task when increasing the number of reasoning trees from 1 to 2. Specifically, in this experiment, FoT's accuracy increased significantly from 56.3% to 77.9% when moving from 1 to 2 reasoning trees, with inference time also increasing accordingly. As the number of reasoning trees continued to increase, the accuracy improvement became more pronounced, particularly when the number of trees increased from 2 to 4, where accuracy further improved to 91.6%. These results demonstrate that in more complex tasks, FoT significantly enhances accuracy by activating multiple reasoning paths and utilizing its internal optimization strategies.
> > >
> > > | **Method**        | **Game24 Accuracy (%)** | **Cost Time (s)** |
> > > |-------------------|-------------------------|-------------------|
> > > | ToT (b=5)         | 56.3                    | 412.2             |
> > > | FoT (n=2, b=5)    | 77.9                    | 571.2             |
> > > | FoT (n=4, b=5)    | 91.6                    | 709.2             |
> > > | FoT (n=8, b=5)    | 96.8                    | 769.2             |
> > >
> > > Moreover, optimization methods such as sparse activation allow the model to effectively select the most promising reasoning paths, keeping computational overhead relatively low while increasing the number of reasoning trees. This demonstrates that FoT performs well in complex tasks, offering substantial accuracy improvements while maintaining efficient inference with manageable computational costs.
> > >
> > > **Q2: The computational cost of FoT.**
> > >
> > > **A2:** Thank you for your comment. As shown in Table 2, although methods like XoT minimize LLM interactions to improve efficiency, they require pre-training on specific tasks for reasoning and later generalization to new problems. This pre-training and generalization process introduces additional computational and time overhead. In contrast, FoT can provide better answers by activating multiple reasoning paths and relying on its own multi-angle deep thinking, without the need for complex pre-training. Notably, when we increase the number of reasoning steps for other methods, they still fail to match FoT's accuracy. For instance, FoT (n=4, b=5) achieves 96.8% accuracy, while BoT (n=8) and XoT (w/ 3 r) (n=8) achieve 83.2% and 87.6% accuracy, respectively.
> > >
> > > | **Method**               | **Average number of LLM calls** | **Success** |
> > > |--------------------------|---------------------------------|-------------|
> > > | IO (n=2)                 | 20.0                            | 10.2%       |
> > > | CoT (n=2)                | 20.0                            | 4.4%        |
> > > | ToT (b=5)                | 13.7                            | 74.0%       |
> > > | ToT (b=8)                | 26.3                            | 78.9%       |
> > > | BoT (n=8)                | 24.0                            | 83.2%       |
> > > | XoT (w/ 3 r) (n=8)       | 31.3                           | 87.6%       |
> > > | FoT (n=3, b=5)           | 23.6                           | 91.6%       |
> > > | FoT (n=4, b=5)           | 25.6                           | 96.8%       |
> > >
> > > This result demonstrates that, while other methods may improve computational efficiency by reducing LLM calls, FoT achieves a significant boost in accuracy. Furthermore, when comparing methods with similar computational costs, FoT consistently outperforms them in accuracy, making it a much more efficient approach. FoT achieves this by avoiding the additional time costs associated with pre-training and generalization, while still delivering superior accuracy.
> > >
> > >
> > >
> > > **Q3: Supplement the details of sparse activation in the main body.**
> > >
> > > **A3:** Thank you for your valuable feedback. We will add more details in Section 3.1 on Sparse Activation to further clarify and strengthen the explanation.

---

### Official Review · Reviewer_zTLg · 2025-03-12

**Overall Recommendation:** 4

**Summary:**

This paper proposes Forest-of-Thought (FoT), a new reasoning framework designed to enhance reasoning ability during test time by combining multiple reasoning trees. This method introduces three strategies (i.e., sparse activation, dynamic self-correction, and consensus-guided decision-making) to enhance both performance and efficiency. Additionally, this paper explores FoT based on two methods, namely ToT and MCTSr. Experiments across various benchmarks, including the Game of 24, GSM8K, and MATH, demonstrate that FoT significantly improves reasoning accuracy and efficiency, validating its effectiveness over existing methods.

**Claims And Evidence:**

Yes. The claims made in the submission are supported by clear and convincing evidence.

**Essential References Not Discussed:**

To the best of my knowledge, there are no essential related works that are missing from the citations or discussion in the paper.

**Experimental Designs Or Analyses:**

Yes, I checked. I think they have no issues.

**Methods And Evaluation Criteria:**

Yes.

**Other Comments Or Suggestions:**

I don't have other comments.

**Other Strengths And Weaknesses:**

Strengths:

(1) This paper introduces a forest structure that integrates multiple reasoning paths by designing sparse activation, dynamic self-correction, and consensus-guided decision-making to enhance the reasoning accuracy, efficiency, and robustness of large language models (LLMs) in complex problem-solving tasks.
(2) This paper explores FoT on two frameworks, i.e., Tree of Thought and Monte Carlo Tree Search, and it shows an improvement compared with baseline.
(3) Experimental results demonstrate the effectiveness of FoT significantly improving the results on Game of 24, GSM8K and MATH.


Weakness:

(1) Please provide a more detailed complexity analysis and comparison of FoT with other methods.
(2) Figure 6 would be more intuitive if presented in a table format.
(3) Table 1 lacks a baseline of FoT, i.e., without self-correction.

**Questions For Authors:**

(1) The authors enhance the input to obtain different starting nodes for the reasoning trees and discuss the effectiveness of this approach in Table 1. However, for the Game of 24, where the input consists of only four simple numbers, how is the input enhanced? Additionally, I am curious about ablation experiments on input enhancement for more complex problems, such as MATH or GSM8K.
(2) Section 3.2 emphasizes that this method integrates pre-defined rules to improve accuracy, but it lacks ablation experiments in this regard. In addition, how are the pre-defined rules defined when testing on the GSM8K and MATH benchmarks？
(3) Table 1 lacks a baseline where FoT is evaluated without the three strategies, making it difficult to assess the importance of self-correction. This makes me wonder whether the improvement primarily comes from the forest structure itself or from the self-correction mechanism

**Relation To Broader Scientific Literature:**

The key contributions of the paper build upon and extend prior work in reasoning frameworks, particularly in relation to Tree-of-Thought (ToT) and Monte Carlo Tree Search (MCTS). The Forest-of-Thought (FoT) approach integrates multiple reasoning trees, enhancing accuracy and efficiency through strategies like sparse activation, dynamic self-correction, and consensus-guided decision-making.


In comparison to ToT and MCTS which focuses on a single tree-based reasoning process, FoT generalizes this idea by incorporating multiple trees to improve robustness and performance.

**Theoretical Claims:**

Yes, I think they have no issues.

---

> ### Author Rebuttal · Authors · 2025-03-31
>
> We sincerely appreciate the thoughtful and comprehensive feedback provided by our reviewers. We will now address the suggestions in detail.
>
> **Q1: Question 1: Data augmentation for the Game of 24 and supplementary experiments on MATH500 and GSM8K.**
>
> **A1:** The Game of 24 input consists of four numbers, and the model is sensitive to their order. Randomly altering the order encourages the model to approach the problem from different perspectives, increasing the number of reasoning paths and improving the accuracy of the solution. By exploring different permutations, the model can consider more possible calculations and solutions. Additionally, we conducted supplementary experiments on the MATH500 and GSM8K datasets. As shown in the table below, input enhancement is equally effective on both the MATH500 and GSM8K tasks.
>
> |               | **MATH500** | **GSM8K** |
> |---------------|-------------|-----------|
> | Without Enhancement | 92.8       | 89.6     |
> | With Enhancement    | 93.4       | 92.1     |
>
> ---
>
> **Q2: Question 2: Ablation experiments on predefined rules in GSM8K and MATH500.**
>
> **A2:** In the math testing tasks such as Math500 and GSM8K, unlike the direct expression computation in Game24, our predefined rules primarily involve checking the correctness of basic operation results added within the text prompts. Therefore, we conducted the following ablation experiments based on Deepseek-R1-Distill-Qwen-7B.
>
> |               | **MATH500** | **GSM8K** |
> |---------------|-------------|-----------|
> | Without Predefined Rules | 92.8       | 89.6     |
> | With Predefined Rules    |  93.1   |     91.7       |
>
> ---
>
> **Q3: Weakness 1: Provide a more detailed complexity analysis and comparison of FoT with other methods.**
>
> **A3:** We conducted multiple rounds of LLM calls for various methods to compare performance. Despite increasing the number of calls, accuracy showed minimal improvement. As shown in the table, even with a similar number of reasoning steps, FoT consistently outperforms other methods in accuracy. For example, FoT (n=4, b=5) achieves 96.8% success, while BoT (n=8) and XoT (w/ 3 r) (n=8) achieve 83.2% and 87.6%, respectively. This shows that, despite more LLM calls, FoT delivers higher accuracy, demonstrating its efficiency in producing quality reasoning results.
>
>
> | **Method**  | **Average number of LLM calls** | **Success** |
> |-----------------------|-----------------|-------------|
> | IO (n=2)   | 20.0       |  10.2%    |
> | CoT (n=2)   | 20.0       |  4.4%    |
> | ToT (b=5)  | 13.7      |74.0%  |
> | ToT (b=8)  | 26.3       |78.9%  |
> | BoT (n=8) |  24.0    | 83.2%   |
> | XoT (w/ 3 r) (n=8) |  31.36    | 87.6%   |
> | FoT (n=3, b=5)   | 23.64      | 91.6%  |
> | FoT (n=4, b=5)   | 25.64       | 96.8%  |
>
> ---
>
>
>
> **Q4: Weakness 2: Replace Figure 6 with a table.**
>
> **A4:** We appreciate the reviewer’s constructive feedback on improving the manuscript. We will revise this into a table format in the final version.
>
> ---
>
> **Q5: Question 3 & Weakness 3: Experimental results of FoT without self-correction in Table 1.**
>
> **A5:** I would like to clarify a point in the 'Results' section of Section 4.2. When FoT does not use the three strategies, it defaults to the Best of N (BoN) method. The experiment starts with BoN, directly applying the ToT framework without input enhancement, sparse activation, or self-correction. In tasks like the Game of 24, based on ToT, the outcome of each step greatly influences subsequent reasoning. Therefore, self-correction at each step is critical. If a basic computational error occurs, further steps become meaningless. We will provide clearer descriptions in the table captions in the final version to explicitly state that the BoN method is equivalent to FoT without the three strategies.
>
> ---

---

### Official Review · Reviewer_Z6BW · 2025-03-14

**Overall Recommendation:** 5

**Summary:**

This paper presents Forest-of-Thought, a reasoning framework designed to enhance the reasoning of LLMs. FoT integrates multiple reasoning trees to leverage collective decision-making for solving complex logical problems. It employs sparse activation strategies to select the most relevant reasoning paths, improving both efficiency and accuracy. Experimental results demonstrate that FoT can significantly enhance the reasoning performance of LLMs.

**Claims And Evidence:**

The paper claims that FoT can enhance the reasoning of LLMs by integrating multiple reasoning trees and employing sparse activation strategies. Figure 1 shows that FoT achieves the 40%+ accuracy gain over ToT on the Game of 24 benchmark.

**Essential References Not Discussed:**

No.

**Experimental Designs Or Analyses:**

The experiments are well organized, evaluating FoT on the diverse benchmarks, comparing against other test-time reasoning (e.g, CoT, ToT, BoN). Additionally, the method is tested on multiple LLM models to demonstrate its generalizability.

**Methods And Evaluation Criteria:**

The paper compares the proposed method against the recent test-time reasoning methods. The evaluation and the results make sense, making it convincing.

**Other Comments Or Suggestions:**

None

**Other Strengths And Weaknesses:**

Strengths:
- FoT offers a new approach for reasoning by integrating multiple reasoning trees, making it can explore diverse reasoning paths and improve decision-making accuracy.
- The proposed sparse activation strategy allows FoT to focus on the most relevant reasoning paths, reducing unnecessary computations and improving efficiency without sacrificing accuracy.
- The framework can be integrated with different LLMs, demonstrating its generalizability across models and datasets.

Weaknesses:
- While FoT improves efficiency through sparse activation, the overall computational cost may still be higher compared to single-path reasoning methods, when it activates multiple btrees. Please clarify this issue.
- I have a concern about implementation. The framework's complexity, involving multiple reasoning trees, dynamic correction, and consensus strategies, could make implementation challenging for other developers.

**Questions For Authors:**

None

**Relation To Broader Scientific Literature:**

The work builds on existing LLMs and tree-based reasoning methods, addressing their limitation of relying a single reasoning path. Deeper discussion with other reasoning frameworks could further clarify its contribution.

**Theoretical Claims:**

There is no theoretical proof in this paper.

---

> ### Author Rebuttal · Authors · 2025-03-29
>
> We are grateful for your time and thoughtful suggestions, which will guide us in improving both the framework and its implementation in future iterations. Below please find the responses to some specific comments.
>
> ---
> **Weakness 1: Computation Efficiency Compared to Single-Path Reasoning.**
>
> **A1:** We appreciate your valuable feedback regarding the computation efficiency of FoT compared to single-path reasoning. While activating multiple subtrees does increase computational overhead, FoT mitigates this by selectively focusing on the most relevant paths, rather than exhaustively exploring all possible paths. Compared to methods that rely on repeated reasoning and averaging results, FoT enhances efficiency by selectively activating only the highest-scoring paths. This selective activation helps maintain or even improve accuracy while reducing computational costs. We will continue optimizing the FoT framework to further improve inference efficiency.
>
> ---
>
> **Weakness 2: Generalization and Use of FoT.**
>
> **A2:** We understand that the complexity of the framework, which includes multiple reasoning trees, dynamic correction, and consensus strategies, may pose implementation challenges for other developers. To mitigate this, we have designed the framework to be modular and well-documented. Furthermore, we are committed to offering continuous support and improving the framework's usability in future updates. With comprehensive documentation and detailed examples, we are confident that developers will be able to implement and adapt the framework efficiently.
>
> ---

---

### Decision · Program_Chairs · 2025-05-01

**Decision:**

Accept (poster)

**Comment:**

This paper proposes Forest-of-Thought, a reasoning framework designed to enhance the reasoning of LLMs for test time. FoT integrates multiple reasoning trees to leverage collective decision-making for solving complex logical problems. It employs sparse activation strategies to select the most relevant reasoning paths, improving both efficiency and accuracy. Experiments across various benchmarks, including the Game of 24, GSM8K, and MATH, demonstrate that FoT significantly improves reasoning accuracy and efficiency, validating its effectiveness over existing methods.

There are four review: three reviewers support accept of this work (two accept, one strong accept). One reviewer insists weak reject with most initial questions are solved. The AC has checked detailed discussion between reviewer and authors and made sure that the details of Sec.3.1 (sparse activation strategy) are clearly stated. In addition, the authors have provided code.

In the final version, the authors should add more explanations on this part.